# Pupillary Light Response Deficits in 4-Week-Old Piglets and Adolescent Children after Low-Velocity Head Rotations and Sports-Related Concussions

**DOI:** 10.3390/biomedicines11020587

**Published:** 2023-02-16

**Authors:** Anna Oeur, Mackenzie Mull, Giancarlo Riccobono, Kristy B. Arbogast, Kenneth J. Ciuffreda, Nabin Joshi, Daniele Fedonni, Christina L. Master, Susan S. Margulies

**Affiliations:** 1Wallace H. Coulter Department of Biomedical Engineering, Emory University and Georgia Institute of Technology, Atlanta, GA 30332, USA; 2Center for Injury Research and Prevention, Children’s Hospital of Philadelphia, Philadelphia, PA 19146, USA; 3Perelman School of Medicine, University of Pennsylvania, Philadelphia, PA 19104, USA; 4College of Optometry, State University of New York, New York, NY 10036, USA; 5Tesseract Health Inc., 530 Old Whitfield St., Guilford, CT 06437, USA; 6Sports Medicine and Performance Center, Children’s Hospital of Philadelphia, Philadelphia, PA 19104, USA

**Keywords:** oculomotor, animal models, mild traumatic brain injury (mTBI), diagnostic methods, pupillary light response, visual processing, porcine

## Abstract

Neurological disorders and traumatic brain injury (TBI) are among the leading causes of death and disability. The pupillary light reflex (PLR) is an emerging diagnostic tool for concussion in humans. We compared PLR obtained with a commercially available pupillometer in the 4 week old piglet model of the adolescent brain subject to rapid nonimpact head rotation (RNR), and in human adolescents with and without sports-related concussion (SRC). The 95% PLR reference ranges (RR, for maximum and minimum pupil diameter, latency, and average and peak constriction velocities) were established in healthy piglets (N = 13), and response reliability was validated in nine additional healthy piglets. PLR assessments were obtained in female piglets allocated to anesthetized sham (N = 10), single (sRNR, N = 13), and repeated (rRNR, N = 14) sagittal low-velocity RNR at pre-injury, as well as days 1, 4, and 7 post injury, and evaluated against RRs. In parallel, we established human PLR RRs in healthy adolescents (both sexes, N = 167) and compared healthy PLR to values obtained <28 days from a SRC (N = 177). In piglets, maximum and minimum diameter deficits were greater in rRNR than sRNR. Alterations peaked on day 1 post sRNR and rRNR, and remained altered at day 4 and 7. In SRC adolescents, the proportion of adolescents within the RR was significantly lower for maximum pupil diameter only (85.8%). We show that PLR deficits may persist in humans and piglets after low-velocity head rotations. Differences in timing of assessment after injury, developmental response to injury, and the number and magnitude of impacts may contribute to the differences observed between species. We conclude that PLR is a feasible, quantifiable involuntary physiological metric of neurological dysfunction in pigs, as well as humans. Healthy PLR porcine and human reference ranges established can be used for neurofunctional assessments after TBI or hypoxic exposures (e.g., stroke, apnea, or cardiac arrest).

## 1. Introduction

Problems with vision are among commonly reported symptoms after a concussion [1], with 70% of children ages 11 to 17 years old reporting at least one problem, typically vergence disorders (the inability of both eyes to track an object up close (converge) or far away (diverge)) and accommodative disorders (the inability to maintain focus on an object as it changes in distance) [2]. Visual deficits affect children’s ability to participate in school-, sports-, and work-related activities and may have long-term impacts on development, learning, and behavior [3].

Clinical pupillary light reflex (PLR) assessments have been incorporated as a diagnostic tool in neurodegenerative diseases such as traumatic brain injury (TBI) [1,4], Alzheimer’s disease [5], Parkinson’s disease [6], and autism [7]. The PLR controls pupil constriction and dilation in response to changes in light intensity to moderate the amount of light that reaches the retina [8]. As an assessment of the integrated response of retinal photoreceptor circuit that is innervated by (via the oculomotor nerve and hypothalamus/brainstem) the parasympathetic and sympathetic pathways to effect constriction of radial and/or circular muscles of the iris [1,9,10], pupillary response evaluations are powerful, noninvasive, and cost-effective assessments of autonomic nervous system function and can be used as quantitative biomarkers for diagnosis of a wide variety of neurological disorders [1,4,6,7]. Typical clinical pupillary dynamics metrics include pupil diameter, pupillary latency (the onset of pupil movement in reaction to the onset of light stimulation), change in pupil diameter, peak pupillary constriction velocity, and peak pupillary dilation velocity [1]. More recent clinical use of the PLR has employed a neurological pupil index (NPi) score derived from a patented algorithm using pupil size, constriction, and dilation parameters [11]. NPi values range from 0 to 5, where scores >3 are healthy and scores <3 are abnormal; this index has demonstrated some clinical utility in predicting TBI patient outcomes for those admitted to hospital [12,13] and identifying patients requiring surgical interventions at triage [14]. The potential to use these assessments in both clinical human settings and preclinical animal models presents a powerful tool for understanding the mechanistic basis for neurological dysfunction.

Thiagarajan and Ciuffreda [15] summarized the human literature and reported that a cohort of military adults with diagnosed concussions had significantly slower constriction and dilation velocity compared with controls when assessed several months after injury. In contrast, high-school-aged children with sports-related concussions (SRC) had opposite findings, where the injured cohorts had greater pupil diameters (maximum and minimum) and constriction velocities (average and peak) than the healthy controls [16,17] when evaluated on average 2 weeks post injury. In addition to the differences in age, the timing of the assessments (chronic vs. acute/subacute) may account for some of these differences.

Animal models provide invaluable opportunities to address the heterogeneity of causation of neurological deficits in humans that may in part lead to the incongruent PLR findings between children and adults at different timepoints after mild TBI (mTBI). Due to the shape of iris muscles, the pupil of mammals such as primates, piglets, and rodents remains circular, similar to humans, at different light intensities, in contrast to domestic cats and seals that have pupils that contract to create a vertical slit or sheep and reindeer that have horizontal pupils [18]. In animals, PLR is typically assessed in the laboratory using devices that are customized for small animals such as mice [19], rats [20], guinea pigs [21], large animals such as dogs [22], and rhesus monkeys [23]. To the authors’ knowledge, there has only been one study examining pupil responses in animals after severe TBI, in which these authors studied head impacts delivered to monkeys and used the presence or absence of the PLR as a confirmation of sustaining a neurological insult [24].

The piglet model is increasingly used as a preclinical model to study a wide range of neurological diseases [25], such as ischemic stroke [26] and Alzheimer’s disease [27], due to the similarities of its morphology and gray- and white-matter distributions that parallel human features [28,29]. The 4 week old piglet is a model of the adolescent brain and has been used to study severe TBI, including diffuse axonal injuries and brain hemorrhages [30,31,32]. In addition, piglet models have been used for mTBI studies for brain biomechanics [33,34], as well as diagnosis [35] and treatment [36]. With particular relevance to this study, an additional advantage of the large animal piglet model is that the conventional clinical equipment used in the clinical setting for humans can be used to study pigs [30], thereby facilitating the translation between species.

The aim of this study was to establish quantifiable PLR biomarkers of mTBI in 4 week old piglets using a clinically available handheld pupillometer designed for humans. First, we evaluated the feasibility of PLR measurements and utility of a pupillometer in 4 week old piglets by establishing a methodology for assessment of pupillary response in a cohort of healthy animals. After confirming feasibility, we established 95% reference ranges (RR) from these healthy animals and validated the RR for each PLR metric with a separate set of healthy animals. Secondly, we used the RR to evaluate a cohort of experimental animals in the acute phase (up to 7 days post injury) that experienced a single or repeated mild sagittal head rotations at load levels scaled from measured human soccer headers in the sagittal direction [37]. We hypothesized that porcine PLR metrics would be affected by load condition (single or repeated) and post-injury timepoint. In parallel, we examined human PLR metrics, using the same assessment tool, in both healthy adolescents and those within 28 days of injury from an SRC. We established similar RR for PLR metrics in the healthy human cohort and examined these metrics in adolescent humans with SRC. Comparison across species allows a more robust understanding of the neurological response.

## 2. Materials and Methods

### 2.1. Piglet Model

Fifty-nine 4 week old Yorkshire piglets (Sus scrofa) were obtained (Palmetto Research Swine, Reevesville, SC and Oak Hill Genetics, Ewing, IL, USA). Animals were housed together on a 7 a.m.–7 p.m. light–dark cycle and permitted to freely eat and drink as desired in the housing pen (LabDiet 5080, St. Louis, MO, USA). Prior to the start of each study cohort, the eyelashes of the animals were trimmed to aid in visualization of the pupil, and animals were acclimated to the behavior study room and commercially available piglet sling restraint with a metal frame for at least 2 days (Lomir Biomedical Inc., Notre-Dame-de-l’Île-Perrot, QC, Canada).

Pupillary dynamics in humans are measured by several technologies such as infrared videography [21], high-speed cameras [38], and smartphones [21]. A number of handheld automated devices, including PLR-200™ and PLRTM-3000 (NeurOptics Inc., Laguna Hills, CA, USA), RAPDx^®^ (Konan Medical Inc., Irvine, CA, USA), and NeuroLight (IDMED, Marseille, France), are commercially available to measure pupillary response [39] and provide quick, quantitative, and repeatable measurements of pupillary dynamics for diagnostic and prognostic purposes. In the current study, one of the handheld infrared pupilometers for humans (PLRTM-3000, NeurOptics, Laguna Hills, CA, USA) was used to capture the pupillary responses to light in piglets [40,41]. The pupillometer (Figure 1) emits a light flash with fixed intensity to stimulate the pupil, captures the pupil via infrared camera (32 frames/s), and determines the pupil diameter with a ±0.03 mm precision. While the top and bottom eyelids were held gently by fingers, the pupillometer eye cup was placed in front of each eye, and the pupilometer was held steady. The eye cup gently encloses the animal’s eye as the pupillometer captures an image of the eye and pupil in response to a light stimulus emitted from the device. A standard positive white-light pulse stimulus, which consists of a bright pulse over a dimmer background that triggers pupil constriction, was used in this study. This white-light stimulus (0.8 s duration, 180 µW) was preceded and followed by a dark (0 µW) background signal. All protocols were approved by Emory University Institutional Animal Care and Use Committee.

On the day of study, animals were fed before 9 a.m., and PLR measurements were performed between 9 a.m. and 1 p.m. Feeding animals before the experiments increased compliance with the assessment. During acclimation and data collection, food rewards, such as dried apple, yogurt chips, and fresh banana, were provided to reward head position and attention to the testing [42,43]. Anesthesia has been known to depress PLR responses such as pupil size, latency, and constriction velocity compared with the pre-anesthesia [22]; therefore, the PLR response of piglets was measured while animals were fully awake and stationary (Figure 2).

The pupillary response was recorded for 5 s to measure the constriction and the subsequent re-dilation of the pupil after the light flash. Repeated measurements of the same eye were performed at least 15 s apart. Figure 1 shows a typical time history of pupillary response to light and illustrates parameters associated with pupillary response. Analysis focused on metrics related to PLR dynamics: latency after light stimulus to constriction initiation (ms), average velocity of pupil diameter constriction (mm/s), and peak velocity of pupil diameter constriction (mm/s). Dilation velocity was not captured consistently and, therefore, was excluded. Abnormal readings were flagged by the device and were not included in data analysis. Additionally, trials with non-negative constriction velocity values, typically associated with spurious pupil diameter readings, were also excluded from the analysis.

### 2.2. Establishing the Healthy Reference Range

From the 59 subjects, we designated a healthy reference cohort (N = 13 females), in which the left and right eyes were tested three times over a 1 week period with ≥3 trials each day. The healthy reference cohort was studied over multiple days to establish the middle 95% healthy reference range (RR) for five metrics: maximum pupil diameter, minimum pupil diameter, pupil latency, average constriction velocity, and peak constriction velocity. An additional cohort of healthy piglets (N = 9, two males and seven females) were allocated to a validation data set. The two male piglets and three female piglets were tested ≥3 times a day for 3 days, and the four female piglets were tested ≥3 times in 1 day, but testing of this set was terminated prematurely due to unplanned facility closures. Therefore, the 19 testing days of this validation cohort was used to evaluate consistency and reliability of the healthy RR generated from the first set.

Because some subjects had usable data from only one eye, we sought to determine if healthy animals’ responses differed between right and left eyes to evaluate the hypothesis that data from one eye could be representative of responses from both eyes to create the 95% RR. Pooling data from healthy reference cohort and healthy validation groups, the results of all trials for each animal were averaged within each day, handling the left and right eye separately. Using paired t-tests (JMP^®^, Version 15. SAS Institute Inc., Cary, NC, USA), we found no significant difference between measurements of the left and right eye (*p* > 0.05).

Next, we evaluated the effect of test day in the healthy animals (one-way repeated-measure ANOVA) and determined no significant effect of test day. Therefore, to determine the healthy RR, each healthy animal’s data were first averaged across trials (within a day), then across the eyes (when only one eye passed quality control, that value was used), and then averaged again across test days, resulting in a single value for each animal. The subsequent values from all 13 animals were again averaged, and the 2.5th and 97.5th percentiles were calculated to generate the 95% RR for healthy animals. Data from each of the validation animals were then compared to the healthy RR to determine the percentage of validation animals that fell within the healthy RR for each PLR metric. For the validation cohort, each animal’s data were first averaged across trials, and then across eyes (when only 1 eye passed quality control, that value was used); next, each day was treated as a separate validation data point resulting in 19 validation values.

### 2.3. Experimental RNR Animals

The remaining (N = 37) animals were allocated to experimental groups: anesthetized shams (N = 10 females); single (sRNR) and repeated, rapid, nonimpact head rotation (rRNR). The sRNR (N = 13 females) consisted of a single sagittal rotation, and the rRNR group (N = 14 females) experienced five sagittal head rotations all within 1 h. The sRNR group experienced a single “high” rotation and the rRNR group experienced one “high” load followed by four “medium” loads spaced 8.4 ± 1.1 min apart (Table 1). The “high” and “medium” levels for piglet TBI loading conditions were scaled from soccer headers in high-school players instrumented with wearable head impact sensors. A total of 267 frontal headers resulting in primarily sagittal head loading were verified via video footage [37]. The kinematic data from the head impact sensors were input into a finite element model of the human brain, and the maximum axonal strain (MAS) was estimated for each head impact [44,45]. From the human data, the 50th and 90th percentile MAS values were identified. These human MAS values were scaled to determine the peak angular velocity and angular acceleration for piglets that would produce a strain-based deformation of the same magnitude [44,45]. Scaled loading conditions for pigs were such that the target load for the single (sRNR) cohort was one scaled 90th percentile “header” (100 rad/s and 36 krad/s^2^) and the repeated (rRNR) cohort target loads were one 90th percentile scaled “header”, followed by four 50th percentile “headers” (60 rad/s and 13 krad/s^2^). From the same human soccer heading data, we found that boys typically experienced six impacts per hour and girls four impacts per hour, both spaced 8 min apart [37].

Head rotations were performed in animals that were sedated with ketamine/xylazine/midazolam (4/2/0.2 mg/kg IM), anesthetized with isoflurane (1–5%) via mask, and administered the analgesic buprenorphine (0.1 mg/kg IM). Rapid nonimpact head rotations (RNR) were delivered using a pneumatic HYGE device (HYGE, Inc., Kittanning, PA, USA) described elsewhere [30]. Angular transducers were affixed to a side-arm linkage to capture angular kinematics of the head rotational events (Applied Technology Associates (ATA), Inc., Arlington, VA, USA and Diversified Technical Systems, Inc., Seal Beach, CA, USA). Measured velocity and calculated accelerations are provided in Table 1. For sham, sRNR and rRNR cohorts, at least three left and right eye PLR measurements were taken on a pre-injury day, as well as on days 1, 4, and 7 post RNR injury.

### 2.4. Human Adolescent Participants

A secondary analysis was performed on PLR data collected from a prospective cohort of adolescents between ages 12 and 19 years with pupillary light reflex (PLR) assessment conducted between 1 August 2017 and 11 May 2021, recruited from a specialty concussion program and private suburban high school, where some results from this cohort have been previously published [16]. The prospective observational cohort study was approved by the Children’s Hospital of Philadelphia institutional review board. Adolescents and/or their parents/legal guardians provided written assent/written informed consent. Pupillary light reflex metrics were measured via the same Neuroptics PLR-3000 handheld, infrared, automated, monocular pupillometer model used for the piglet portion of this study [46]. The pupillometer is approved by the US Food and Drug Administration and has been used in similar studies of mTBI in human adults and adolescents. The diagnosis of SRC was made by trained sports medicine pediatricians on the basis of the most recent Consensus Statement on Concussion in Sports [47], and all adolescents with concussion were assessed with pupillometry within 28 days of injury. Overall, PLR data from 167 healthy controls were used to establish a healthy RR for humans. PLR metrics for 177 concussed cases were obtained and compared to the healthy RR.

### 2.5. Statistics

For each PLR metric, comparisons between the experimental animals and the healthy RR were completed. Data for each injury group (sham, sRNR, and rRNR) and study day (preinjury, as well as days 1, 4, and 7 post injury) were evaluated to determine the proportion of animals that fell within the healthy RR. To test if the proportion of animals that fell within the healthy RR on each day was significantly different by loading group (group effect at each day), a Fisher’s exact test was conducted. To test if the proportion of injured animals that fell within the healthy RR for each loading group differed by day (day effect per loading group), a Cochran’s Q test was performed followed by a McNemar’s post hoc test within each injury group.

We did not assume that the effects of sagittal loading on the visual pathway were axisymmetric. Therefore, a three-way ANOVA with repeated measures for day was run to test the effect of eye (left or right), day, and loading group (sham, sRNR, and rRNR). For metrics where eye was not a significant factor, a two-way ANOVA with repeated measures was performed to determine the effect of study day and loading group (sham, sRNR, and rRNR) with Bonferroni post hoc tests for each PLR measure. All statistical analyses were conducted using SPSS Statistical software (V 28, IBM), and significance was accepted at *p* < 0.05.

For human PLR metrics, the proportion of concussed cases with PLR metrics within the healthy RR was compared to the proportion of healthy controls within the healthy reference range . The proportion of PLR metrics from cases at initial visit that were within the healthy reference range were compared to healthy controls using χ^2^ with a significance level of 0.05. Additionally, the proportion of concussed cases with PLR metrics within health RR was compared by sex and previous history of concussion using χ^2^ with a significance level of 0.05.

## 3. Results

### 3.1. Animal Results

For each of the five PLR metrics, 95% of the validation animals fell within the healthy RR for maximum diameter and latency, 84% of validation animals fell within the RR for minimum diameter, and 68% of validation animals fell within the RR for average constriction velocity. However, 0% of validation animals fell within the RR for peak constriction velocity; thus, peak constriction velocity was excluded from further analysis, resulting in statistical comparisons completed for the remaining four metrics (maximum and minimum diameter, latency, and average constriction velocity). The values for mean, standard deviation, 2.5th, 50th, and 97.5th percentiles of pupillary response metrics from healthy animals (N = 13) are provided in the Appendix A
Table A1. Maximum and minimum diameter were the only metrics that had significant findings for the reference range statistical analysis, where day 1 post RNR responses were significantly below the healthy RR for rRNR compared with shams and their own pre-injury baselines (Figure 3A,C).

The three-way ANOVA did not yield significant results for maximum diameter, minimum diameter, and latency; therefore a two-way ANOVA was conducted for these metrics. Two-way repeated-measure ANOVA results demonstrate that sRNR had reduced maximum diameters at days 1 and 7 post TBI in comparison to pre-injury levels (Figure 3B). The rRNR group had reduced maximum diameters relative to pre-injury on all days studied, where values on day 1 post TBI were significantly smaller than those on days 4 and 7 (Figure 3B). There was a main effect of load group on minimum diameter on day 1 post TBI, where the rRNR had reduced minimum diameters compared to sham (*p* = 0.025). Furthermore, sRNR animals had significantly reduced minimum diameters at day 1 relative to pre-injury levels, where day 1 remained significantly reduced compared to day 4 and 7 (Figure 3D). The rRNR animals had significantly reduced minimum diameters on all post-injury days relative to pre-injury, where day 1 values remained significantly lower than days 4 and 7 (Figure 3D). There were no significant findings for the reference range analysis or two-way ANOVA findings for latency (Figure 3E,F).

For average constriction velocity, the three-way ANOVA showed a significant interaction effect between side and loading group. Figure 4A illustrates the average constriction velocities for both the left eye (solid dot) and right eye (patterned dot) of each animal per loading group (sham, sRNR, and rRNR). Fisher’s exact test for the effect of group at each day was significant for the left eye on all days; however, results for the right eye did not reach significance. Because the Cochran’s Q test and McNemar’s test were not significant for either eye, we found no overall or day-by-day relationship between groups for the proportion of animals within the average constriction velocity healthy RR. Figure 4B shows the post hoc tests from the ANOVA that employed paired t-tests for eye per loading group, and it was found that the left eye had faster constriction velocities than the right eye in the sham group at pre-injury and in the sham and sRNR groups at day 1 (*p* < 0.05). Furthermore, significant differences were found for right eyes in the single group between pre-injury and day 1, for right eyes between single and repeated RNR groups on day 1, and for left eyes between sham and sRNR on day 4 (Figure 4B).

### 3.2. Human Results

The patient data and patient characteristics for the human adolescent cohort of healthy and concussed individuals is described in Table 2 and Table 3.

Overall, PLR data from 167 healthy controls were used to establish a healthy RR for humans (Table A2 in Appendix A). The 95% reference ranges for latency (180.0–243.3 ms), average constriction velocity (1.26–3.91 mm/s), peak constriction velocity (2.00–6.21 mm/s), maximum pupil diameter (2.67–5.57 mm), and minimum pupil diameter (2.02–3.4 mm) were established from the PLR metrics from the sub-cohort of healthy adolescents.

PLR metrics for 177 concussed cases were obtained and compared to the healthy RR (Table 2). Adolescents with concussion had a significantly lower proportion (85.8%) within the healthy reference range for maximum pupil diameter compared to healthy adolescents (95.8%). The proportions of all other PLR metrics within the healthy reference ranges for concussed did not have significant differences from the proportion of healthy adolescents within the reference range compared to the corresponding PLR metrics.

In contrast to the piglet data, there was no difference in PLR metrics between adolescents with multiple concussions compared to those with only one concussion. In addition, there was no significant difference in PLR metrics between female and male adolescents (see Table A3 and Table A4 in Appendix A).

## 4. Discussion

This is a first report of the pupillary light reflex in a porcine preclinical model of TBI. Importantly, a commercially available pupillometer, commonly used in clinical assessments, could obtain reliable and consistent PLR measurements in 4 week old piglets to obtain maximum and minimum pupil diameter, highlighting interesting nuances with average constriction velocity. We established healthy and validated reference ranges which can serve as baseline comparisons for use in the preclinical and animal sciences literature. We show that head rotations, characteristic of typical heading kinematics in soccer, yielded significantly reduced maximum and minimum diameter on day 1 post sRNR in comparison to pre-injury. In addition, the rRNR group had faster average constriction velocities than the sRNR group on day 1 post injury. Interestingly, pupil diameters (maximum and minimum) tended to approach recovery on days 4 and 7; however, these metrics remained significantly lower than at pre-injury, which suggests that PLR deficits may persist beyond the acute (7 day) period for mTBI in piglets. The injury paradigm employed in the animal model illustrated the effect of severity when comparing a single head rotation to repeated head rotations. There were a greater number of significant pupil deficits in the rRNR group than the sRNR group, as well as a greater magnitude of those differences on each study day (e.g., maximum and minimum diameter, Figure 3A,B and Figure 3C,D).

In addition, to compare across species, we examined human adolescent PLR metrics, from both healthy and concussed individuals (Figure 5). We established an RR from healthy adolescents for each PLR metric and compared the proportion of concussed and healthy adolescents falling within the RR for each group. In our investigation of the human adolescent PLR data, we found that maximum pupil diameter was the only PLR metric to have a significantly fewer concussed adolescents falling within the RR, whereas the other metrics yielded similar proportions of healthy and concussed adolescents within the RR. This may be, in part, due to the mild nature of SRC along the spectrum of TBI, likely representing the mildest form of mTBI and, as such, resulting in subtle physiological perturbations. Quantitative PLR metrics remain a promising target outcome measure for identifying and monitoring SRC; our findings in this investigation indicate that a narrower RR with more stringent criteria for “healthy” may be useful, and future studies should investigate optimizing sensitivity of these measures to an mTBI such as SRC, while maximizing specificity.

Interestingly, our comparative study of piglet and human PLR metrics revealed differences in the direction of post-mTBI alterations. Specifically, while the piglets PLR metrics were diminished acutely (during the first week) after mTBI, the PLR metrics in the concussed adolescents were enhanced sub-acutely (<28 days) after mTBI. The athletic adolescent population may be a special cohort in comparison with their nonathletic counterparts, potentially exhibiting unique pupillary responses as a result of sports and skill demands. With the assumption that the developmental stages of the piglet and human adolescents are similar, the main remaining explanation for the different findings could be the timing of assessment and similarity of the injury severity and mechanism (total number, direction of head loading, and magnitude).

In a study examining 200 adult military personnel in the acute phase of concussion (<72 h) after non-blast-related mTBI (i.e., from aerial jumps, motor vehicle accidents, falls, sports, and recreation), the authors found that those with an mTBI had the following decreased PLR metrics: maximum and minimum diameter, percentage constriction, average constriction velocity, and average dilation velocity in comparison to control (N = 100) [48]. A separate study investigated military patients diagnosed with blast-induced mTBI (N = 20) compared to age-matched controls (N = 20) in the subacute period post injury (15–45 days), and the results also reported that maximum and minimum pupil diameters, average constriction velocity, and maximum constriction velocity were significantly lower in the mTBI group [49]. Another report in military subjects with chronic (<1 year) non-blast mTBI (N = 17) compared to healthy controls (N = 15) also found decreased maximum constriction velocity, average constriction velocity, and maximum and minimum pupil diameter among others [46]. These acute and chronic results in adults are similar to our piglet results.

Age has a demonstrable influence on injury responses. In contrast to adults, we (Master and Podolak [16]) examined a younger human population and conducted a study of high-school athletes (12–18 years old) with a sports-related concussion (N = 110) and measured PLR in the subacute period (within 28 days). We found that the injured cohorts had greater pupil diameters (maximum and minimum) and constriction velocities (average and peak) than the healthy controls (N = 143). We hypothesized that concussion may cause a traumatically induced autonomic dysfunction in the sympathetic pathway, the primary driver influencing the dilation PLR response. Greater maximum pupil diameters allow more light to enter the eye during light stimulation, affecting subsequent constriction velocity and dilation dynamics [16]. In a separate study, Hsu and Stec [17] obtained PLR in children seen in a concussion clinic (N = 92) with an average date from injury of almost 2 months and reported similar findings, where average and peak constriction velocities were greater in the concussed pediatric cohort than in controls (N = 192).

Differences in these PLR findings between adults and children have been attributed to age as PLR measures have been known to be sensitive to age and change throughout development [1,21]. Additionally, the time post TBI when the PLR measurements were taken represents another factor that could lead to incongruous findings. Lastly, the mechanism of injury is an additional potential contributing factor for the differences observed. Pupil diameters (minimum and maximum) were not found to be different between groups of children in Hsu and Stec’s [17] study; however, the values were larger (6 mm) than those reported in Master and Podolak’s [16] study (4.8 mm). A larger maximum pupil diameter at the beginning of the PLR test would result in larger response amplitudes, as well as greater pupil dynamics and constriction velocities (average and peak), which could help to explain the findings observed in these pediatric studies [15]. Future longitudinal studies are needed in order to delineate the trajectory of PLR metrics over the clinical course after injury, to determine the pattern of changes in adults and children, which may better characterize a pattern with diminished PLR metrics both acutely and chronically, perhaps demonstrating a subacute enhancement of PLR measures in the first weeks after injury.

Injury setting may contribute to the differences in PLR responses, in addition to age and timing. Studies involving military personnel had subjects with concussions from blast injury and blunt force head trauma from aerial maneuvers, falls, and combat training. In contrast, the pediatric studies were primarily injuries due to sport and recreation (falls onto various surfaces and player-to-player contact). The head loading conditions, comprising various combinations of surfaces, inbound velocities, impact locations, and directions, may play an important role in how energy is translated to the head and brain resulting in trauma. Variations in loading conditions have been demonstrated to result in distinct patterns of head kinematics [50], stresses and strains from computational modelling [51], neurobehaviors in animals [52], and histopathology [51]. The studies involving concussions in children and adults did not control for biomechanical differences in head loading conditions, which may further contribute to the disparate findings.

Our piglet study had several limitations. First, we studied head loading in the sagittal direction only, which applies the biomechanical loads equally across the two hemispheres and visual pathways. Despite restricting the study to the sagittal direction to limit the influences of asymmetric kinematics on the eyes, we observed differences between the left and right eyes, albeit only for average constriction velocity and in the sham and sRNR groups (Figure 4D). Other directions may reveal more pronounced sidedness. Most studies in the literature treated patient PLR data as an average between the left and right eye [16,17]; however, in a more recent clinical study involving stroke and TBI patients, a difference in the Npi score of the right and left eye was associated with poorer outcomes and greater levels of disability at discharge [53]. Therefore, future studies changing loading direction should continue to conduct analyses specific to the left or right eye. Second, only female piglets were examined in this study for the injury cohort, which may limit extrapolations to male piglets. In a study examining males and females and tracking the time to recovery after a concussion, females were found to take longer to recover, with a longer time for symptoms to abate and to return to scholastic and sports activities than males [54]. Therefore, the trajectory of PLR deficits and recovery may not only be sex-dependent but also affected by age, as well as injury severity and mechanism. Future work should include changing head loading direction, including male animals, and studying animals beyond 7 days to track when PLR deficits return to baseline to better understand injury severity (single and repeated RNR) and predict recovery trajectories on the basis of biomechanical conditions.

## 5. Conclusions

In this study, we obtained consistent and reliable pupillary light reflex responses in 4 week old piglets using a commercially available pupillometer commonly used in clinical assessments and established healthy reference ranges for a substantial number of animals. Although this study highlighted some species differences in pupillary light metrics, conducting parallel experimental studies using the same device, methodology, and metrics in human and animals, as introduced in this study, can facilitate preclinical–clinical translation of objective, involuntary diagnostics and treatment efficacy metrics. We applied these ranges to study PLR in animals subject to single and repeated mild head rotations characteristic of non-injurious heading in soccer during the acute time period (within 7 days); we found that the most severe deficits occurred on day 1 after injury, and that repeated injuries tended to be more severe than a single mild head rotation as reflected in a greater number of PLR deficits and a larger magnitude of those decreases. Furthermore, pupil diameters (maximum and minimum) remained significantly decreased 7 days after low-velocity head rotations. We conclude that PLR holds promise as a feasible involuntary quantitative physiological metric of neurological dysfunction in piglets.

## Figures and Tables

**Figure 1 biomedicines-11-00587-f001:**
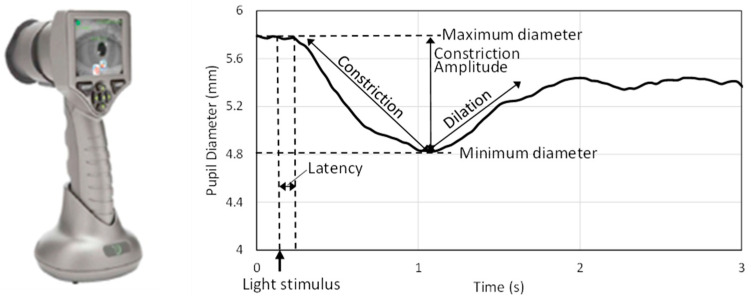
Handheld PLR-3000 device (**left**) and a typical pupil diameter time history and pupillary response parameters (**right**).

**Figure 2 biomedicines-11-00587-f002:**
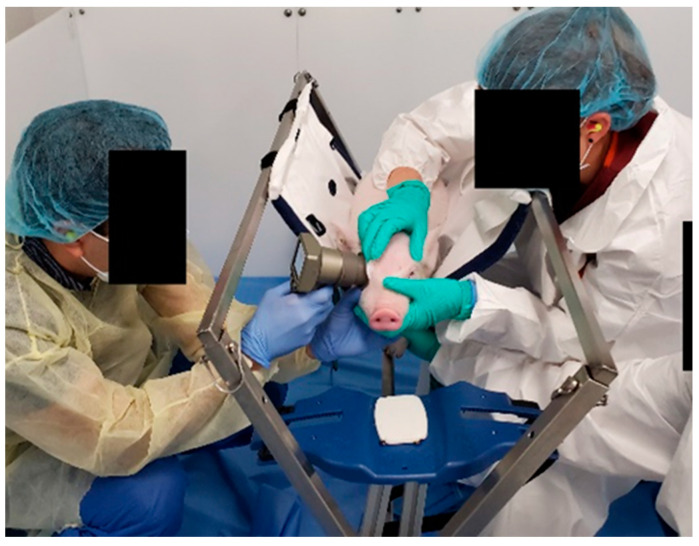
The experimental setup used in this study showing the position of the piglets in the sling and placement of the pupillometer on the animal’s eye.

**Figure 3 biomedicines-11-00587-f003:**
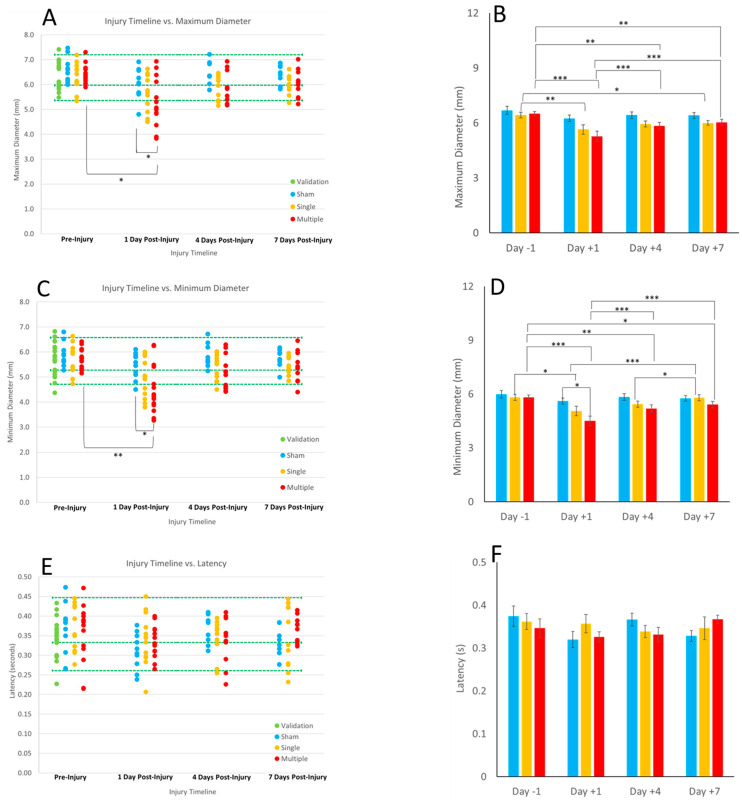
Proportion of validation and experimental animals in the healthy reference range (**A**,**C**,**E**) and two-way repeated-measure ANOVA results for the effect of injury group and study day (**B**,**D**,**F**) for maximum diameter (**top row**), minimum diameter (**middle row**), and latency (**bottom row**). Blue bars represents sham, yellow bars represents sRNR, and red bars represents rRNR groups. The healthy RR is demarcated with green dotted lines in (**A**,**C**,**E**). Overlaying cross bars show significant comparisons with * *p* < 0.05, ** *p* < 0.01, and *** *p* < 0.001.

**Figure 4 biomedicines-11-00587-f004:**
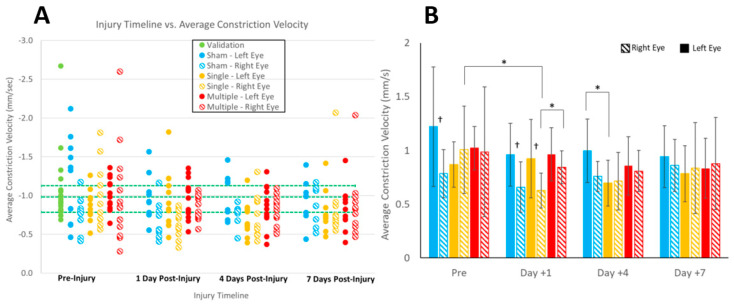
Proportion of validation and experimental animals in the healthy reference range (**A**) and three-way repeated-measure ANOVA for side, injury group, and study day on average constriction velocity (**B**). The effect of side was found to be significant for average constriction velocity; therefore, left and right eye data are plotted (**B**). Blue bars represents sham, yellow bars represents sRNR, and red bars represents rRNR groups. The healthy RR is demarcated with green dotted lines in (**A**). Overlaying cross bars show significant comparisons with * *p* < 0.05. † in (B) illustrates significant differences between left and right eyes (*p* < 0.05).

**Figure 5 biomedicines-11-00587-f005:**
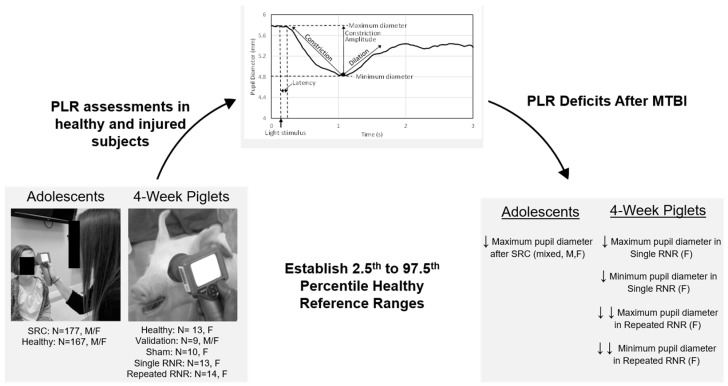
Illustration of PLR cohorts and comparison of main findings for adolescent children with and without an SRC and healthy and experimental RNR 4 week old piglets. M = male, F = female. Mixed refers to mixed head injury mechanisms causing the SRC. A greater number of PLR deficits were observed in controlled piglet RNR studies, with repeated RNR having more severe deficits.

**Table 1 biomedicines-11-00587-t001:** Mean ± standard error of angular velocity and angular accelerations for single and multiple RNR loading groups.

Injury Group	Load Level	Angular Velocity (rad/s)	Angular Acceleration (rad/s^2^)
Single	High	104.3 (±0.49)	38,444 (±1272)
Multiple	Medium	61.18 (±0.20)	15,033 (±169.0)
	High	104.5 (±0.40)	38,394 (±509.3)

**Table 2 biomedicines-11-00587-t002:** Percentage within 2.5th and 97.5th percentile reference range comparison for human study cohort.

Metric	Cases at Initial Visit (N = 177)
Latency (ms)	94.9
Average constriction velocity (mm/s)	92.6
Maximum constriction velocity (mm/s)	93.2
Maximum pupil diameter (mm)	85.8 *
Minimum pupil diameter (mm)	99.4

*Adolescents with concussion had a significantly lower proportion within the healthy reference range for this parameter compared to healthy adolescents.

**Table 3 biomedicines-11-00587-t003:** Demographic and clinical characteristics of the study cohort.

Demographics	Healthy Controls (N = 167)	Concussed Cases (N = 177)
Sex		
Female	91 (54.5%)	97 (54.8%)
Male	76 (45.5%)	80 (45.2%)
Race/ethnicity		
Non-Hispanic White	125 (77.6%)	144 (81.8%)
Non-Hispanic Black	19 (11.8%)	14 (7.9%)
Hispanic	6 (3.7%)	6 (3.4%)
Non-Hispanic Asian	5 (3.1%)	3 (1.7%)
Non-Hispanic mixed race	5 (3.1%)	1 (0.6%)
Non-Hispanic other	1 (0.6%)	8 (4.6%)
Age (Median, IQR)	15.6 (14.3–17.2)	15.4 (14.2–16.6)
Previous concussion	40 (23.95)	81 (45.76)

## Data Availability

The datasets used and/or analyzed during the current study are available from the corresponding author on reasonable request.

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
