# Peer review of "Pupillary Light Response Deficits in 4-Week-Old Piglets and Adolescent Children after Low-Velocity Head Rotations and Sports-Related Concussions"

_biomedicines, 2023, doi:10.3390/biomedicines11020587_

Round 1
Reviewer 1 Report
In the present paper, the authors aimed to establish quantifiable pupillary light response (PLR) biomarkers of mild traumatic brain injury (mTBI), also referred to as concussion, in 4-week-old piglets using a clinically available hand-held pupillometer designed for humans, in the acute phase, ie.e up to 7 days post-injury. Additionally, examined human PLR metrics, using the same assessment tool, in both healthy adolescents and those within 28 days of injury from sports-related concussion (SRC).
As the authors state themselves, it is important to establish and evaluate the potential to use these assessments in both human clinical settings and pre-clinical animal models because it might prove to be a powerful tool for understanding the mechanistic basis for neurological dysfunction. Therefore, there is value to the results presented in this manuscript.
However, some major and minor issues need to be addressed.
1. What is most lacking in the paper is the missing information regarding human studies. Namely, as the authors stated, part of the results from the cohort of patients used in this study was already published previously (Master et al., 2020). In that paper, unlike in this one, additional information regarding concussion patients was presented that I believe is necessary to be included in this paper also. This is particularly related to the history of previous concussions in patients, and largely because in the animal part of the study, some differences between the single mTBI and repeated mTBI were noted.
2. In the results from the human study, Table 3 seems to be wrong, as the results in the text and the table regarding pupil diameter do not match. In the table, it says “baseline” vs. the text where it says that it is the “maximum” pupil diameter that is present in the significantly lower proportion of concussion patients.
3. The authors addressed some of the possible limitations of the study. One of them is related to only using female piglets in the animal part of the study. Since in the human study patients of both sexes were included, it would have been interesting to see, since the data already exists, if there are differences present in those subjects.
4. The title of the paper is very detailed, but the problem is that it only relates to the results of the animal study. Authors should find a way to include also the results from human studies.
5. Abstract should be rewritten. The first sentence is wrong. Namely, neurological disorders and TBI are ONE OF leading causes of death and disability, but not A leading cause (it is still heart disease). Also, it says that the used model was done in a “pre-adolescent porcine model”, and on page 2, line 91, it says “4 week-old piglet is a model of the adolescent brain”. In regards to sex, please add that only female piglets were used also in the abstract, and in the human studies, it is both sexes.
6. On page 5, line 191, the text goes: “determined no significant effect of test day (p<0.05)”. If p<0.05, that usually signifies significance.
7. Table legends for Tables 2 and 3 are lacking. For example, instead of just “Demographics” for Table 2., maybe write (as in the previous paper): “Demographic and Clinical Characteristics of the Study Cohort”.
8. Figure quality in the document provided for the revision is very bad so I suggest authors provide better quality images before publication.
9. References need to be updated, in my opinion, as there is e.g. only one from 2022. I think the authors should also include findings from a study by Privitera et al. BMC Neurol. 2022;22(1):273 as it is relevant to the differences in pupillary response between the left and right eye in the brain injury, something that the authors also discovered in their research.
In conclusion, this manuscript could be acceptable for publication upon the correction of the mentioned issues.
Reviewer 2 Report
The MS "Decreased Pupillary Light Responses in 4-Week-Old Piglets After Single and Repeated Low Velocity Head Rotations in Comparison to Healthy Reference Ranges" is well written and really clear. The MS has no major flaws.
The hypothesis and aims are clear stated, as well as the limitations. Authors clearly describe the methods.
Minor:
I would suggest to include a figure summarizing the finding like a graphical abstract it would help to understand. Also I would suggest to hypothesized/develop on the causes of such differences in the discussion section.
Major: I don't know what happened with the figures, but I am not able to read any info. They are blurred. Could you please upload figures with an higher resolution?
Round 2
Reviewer 2 Report
Thanks for the changes